# Effects of Cold Atmospheric Plasma Pre-Treatment of Titanium on the Biological Activity of Primary Human Gingival Fibroblasts

**DOI:** 10.3390/biomedicines11041185

**Published:** 2023-04-16

**Authors:** Madline P. Gund, Jusef Naim, Antje Lehmann, Matthias Hannig, Constanze Linsenmann, Axel Schindler, Stefan Rupf

**Affiliations:** 1Clinic of Operative Dentistry, Periodontology and Preventive Dentistry, Saarland University, 66421 Homburg, Germany; 2Leibniz Institute of Surface Modification (IOM), 04318 Leipzig, Germany; 3ADMEDES GmbH, 75179 Pforzheim, Germany; 4Piloto Consulting Ion Beam and Plasma Technologies, 04668 Grimma, Germany; 5Synoptic Dentistry, Saarland University, 66421 Homburg, Germany

**Keywords:** biological cell activity, cell attachment, cold atmospheric plasma, primary human gingival fibroblasts

## Abstract

Cold atmospheric plasma treatment (CAP) enables the contactless modification of titanium. This study aimed to investigate the attachment of primary human gingival fibroblasts on titanium. Machined and microstructured titanium discs were exposed to cold atmospheric plasma, followed by the application of primary human gingival fibroblasts onto the disc. The fibroblast cultures were analyzed by fluorescence, scanning electron microscopy and cell-biological tests. The treated titanium displayed a more homogeneous and denser fibroblast coverage, while its biological behavior was not altered. This study demonstrated for the first time the beneficial effect of CAP treatment on the initial attachment of primary human gingival fibroblasts on titanium. The results support the application of CAP in the context of pre-implantation conditioning, as well as of peri-implant disease treatment.

## 1. Introduction

Titanium provides excellent biocompatibility, a high flexural strength, and corrosion resistance. Therefore, it is widely used in medicine, particularly in the field of implantology, as well as for osteosynthesis plates, or bone screws in all fields of surgery. In dentistry, titanium is the most used implant material of choice for replacement of missing teeth [1]. Various factors, such as wettability or hydrophilicity, influence cell behavior and attachment [2,3]. Technical methods such as sandblasting, etching and coating are used to modify surface properties, aiming to improve the adhesion of the surrounding tissue to the titanium implant and promoting an early adhesion of osteoblasts and osteosynthesis [4,5,6]. Furthermore, a dense attachment of gingival fibroblasts and epithelia cells is essential for successful implant therapy [7]. The peri-implant tissue establishes a barrier and protects against colonization by bacteria of the oral environment [8]. Thus, colonization of the implant surface with microorganisms and inflammation of peri-implant bone can be prevented [9]. Nevertheless, the excellent biocompatibility of titanium also allows the attachment of microorganisms and therefore the adsorption of biofilms on the implant surface [10]. Microbial biofilms are playing the decisive role in the development of peri-implant diseases such as perimucositis and peri-implantitis [11]. Again, fostering human cell adhesion on the implant surface is essential. However, different cell species have a variable interaction with the titanium surface. Fibroblasts and epithelia cells show the best adhesion on machined surfaces, while osteoblasts favor microstructured surfaces [12,13].

Cold atmospheric plasma (CAP) can be used for the removal of biofilms [14,15,16]. CAP also enables a contactless surface modification, in particular, fissures and cavities, leading to an improved wettability, also on dental implant materials [17,18]. There are already studies available that investigated cell behavior and adhesion patterns of human gingival fibroblasts (HGF-1) and osteoblast-like cells (MG-63) on coated and uncoated titanium and on zirconia discs after cold plasma treatment, pointing out the faster proliferation and adhesion of cells from surrounding tissue [19].

The aim of this study was to investigate the attachment and the biological activity of primary human gingival fibroblasts on machined and microstructured titanium after CAP modification.

## 2. Materials and Methods

Titanium discs: A total of 120 titanium discs (titanium grade 2, Friadent, Mannheim, Germany, diameter 5 mm or 10 mm, height 1 or 2 mm) with machined and microstructured (sand-blasted, air-abraded, roughness of 2 µm, mean maximum value of the tread depth of 21.35 µm) surfaces were used for test specimen. A total of 84 titanium discs were treated, with 36 samples being treated with the gas flow without ignition of the plasma. Titanium discs with a diameter of 5 mm were utilized for the fluorescence microscopic evaluation and with a diameter of 10 mm for scanning electron microscopic examination.

Plasma Source: A plasma source of the Leibniz Institute for surface modification was used for the plasma treatment, with the plasma jet being generated by a pulsed microwave (2.45 GHz). The plasma treatment parameters were adjustable in terms of pulse power, pulse width, and mean microwave power. The plasma source was mounted on a computer-controlled 3-axis movement system (Steinmeyer MC-G047, Feinmess Dresden GmbH, Dresden, Germany). The chosen processing parameters—5 µs pulse width and 300 W peak power—resulted in a mean power of 5 W and a scan velocity of 8 mm/s. The surface was processed by means of meander-like line scans through the plasma jet. The working distance was 2 mm, and the gas flow was 2 L/min, made up of two different gas mixtures. The noble gas Helium was chosen as carrier gas. The gas mixture contained 2000 sccm helium and 5 sccm oxygen.

Primary human gingival fibroblasts: Primary human gingival fibroblasts were obtained during osteotomy of third molars in the Department of Oral and Maxillofacial Surgery, Saarland University Hospital, Homburg. The subsequent cultivation took place under constant conditions (37 °C, 5% CO_2_, 2 times per week change of medium), cells of the sixth passage were selected for the series of experiments. The use of the cells was approved by the Ethics Committee of the Saarland Medical Association (E168/09), and the donors gave their written consent.

Sample treatment: The samples were treated under stable conditions in an S2-laboratory (air temperature: 22 ± 2 °C, eightfold air change per hour, light overpressure). The application of gingival fibroblasts was performed immediately after plasma treatment in a well plate under an aseptic bench. The 10^4^ cells were applied on treated samples and the non-treated controls. The titanium samples were constantly kept moist over a period of 30 min with DMEM liquid medium before the cavities were filled with 200 µL DMEM liquid medium. Afterwards, cultivation was performed in an incubator (Hera Cell 240, Thermo Scientific, Waltham, MA, USA) at 37 °C and 5% CO_2_ for 4, 12 and 24 h.

Sample analysis: The analysis of the attachment of gingival fibroblasts was performed by fluorescence microscopy. Vinculin, Phalloidin and DAPI staining were used. The fixation of cells on specimens was performed using 2.5% glutaraldehyde (Carl Roth GmbH, Karlsruhe, Germany) for 30 min. To remove the glutaraldehyde, the specimens were rinsed five times with PBS, followed by permeabilization of the cell wall with 0.1% Triton-X (Roche, Grenzach-Wyhlen, Germany) in PBS. Thereafter, the samples were washed two times by PBS, the nonspecific background being reduced using 1 % BSA (bovine serum albumin in PBS, Gibco, Invitrogen, Carlsbad, CA, USA). Afterwards, the specimens were incubated in a humid chamber with a primary antibody against vinculin (Monoclonaler Anti-Vinculin, antibodies of mice; Sigma-Aldrich Chemie GmbH, Steinheim, Germany) being dissolved in 5% BSA, followed by a double lavation with 0.1% Tween 20 (Roche, Grenzach-Wyhlen, Germany) in PBS and removal of the Tween 20 by PBS. Subsequently, the specimens were washed twice with 0.1% Tween 20 (Roche, Grenzach-Wyhlen, Germany) in PBS, followed by the removal of Tween 20 in PBS. After incubation, the specimens graduated with a secondary antibody (anti-mouse IgG, RD Systems, Minneapolis, MN, USA) for one hour in a humidified chamber; re-washing in 0.1% Tween 20 and a subsequent washing by PBS followed. Next, the incubation concluded with a phalloidin stock solution (1 mg/mL MetOH) for 25 min, in the dark, at room temperature. This was followed by the application of DAPI solution (Roche, Grenzach-Wyhlen, Germany) for ten minutes, at room temperature, in the dark.

The titanium disks were mounted on slides (Superfrost; Menzel-glasses, Braunschweig, Germany) and evaluated under a reflected-light microscope (Axio Scope A1, Carl Zeiss, Göttingen, Germany). Recordings were made with lenses of 2.5× magnification, 10× and 20×. The recordings at a magnification of 25 (2.5 (lens) × 10 (ocular)) were used for each sample to create a general overview, as well as to evaluate the percentage of cell cover. Five shots were customized at 100× magnification, each from the center of the sample and from the border area. These images were used for analyzing the number of cells, the attachment behavior and the average nucleus size. AxioVision 4.8. software (Carl Zeiss Microlmaging, Göttingen, Germany) was used to evaluate attachment, average nucleus size and number of cells. The scanning electron microscopy examination was performed with a FEI XL 30 ESEM FEG (Fei, Eindhoven, the Netherlands). A total of 36 treated titanium discs and 16 untreated controls were analyzed. These were fixed in 2.5% glutaraldehyde for 30 min, rinsed five times for ten minutes with PBS and followed by 4% osmic acid (osmium tetroxide, Carl Roth GmbH, Karlsruhe, Germany), and then they were fixed the second time for 20 min. This was followed by five additional washes of ten minutes each with PBS, dehydration with an ascending alcohol series and 1,1,1,3,3,3-hexamethyldisilazane (HMDS, Merck AG, Darmstadt, Germany). The samples were applied on SEM stubs (Plano, Wetzlar, Germany) and coated with a 2–3 nm thick platinum layer (Sputter Coater SC 7640, Quorum Technologies Ltd., Lewes, UK). Each sample was then investigated under the scanning electron microscope, under three levels of magnification (radiograph 25-, 500- and 1000-fold). The overview screen was used for the evaluation of cell coverage on the test specimens. In some cases, higher or lower magnifications were used (200- to 8000-fold) to obtain images of the cells. The scanning electron microscopic examination was used to verify the fluorescence microscopic analysis, allowing statements on the phenotypic formation of the fibroblast cell body.

The colorimetric determination of alkaline phosphatase activity was performed by using a kit (Biovison, Mountain View, CA, USA). A volume of 80 µL medium was taken from each well of the culture plates containing titanium discs and transferred into a 96-well plate (Greiner bio-one, Frickenhausen, Germany). Subsequently, 50 µL of pNPP solution was added, and the plate was then left for 60 min, in darkness, at room temperature. Simultaneously, a color sample (blank) was prepared. As a standard, 40 µL of 5 mM pNPP solution was used in 160 µL of assay buffer. A series of ascending concentrations was prepared for the standard curve. All experiments were performed twice. After one hour, the reaction was interrupted using 20 µL stop solution to measure the absorbance in the ELISA reader (Tecan Infinite 200, Magellan V6.6, Tecan, Groedig, Austria) at a wavelength of 405 nm.

After cultivation for 4, 12 and 24 h, 20 µL of WST-1 reagent (Cell proliferation kit WST-1, Water soluble tetrazolium) was added to every cavity of the culture plates. In addition, a control well containing pure medium without WST-1 reagent was prepared, and a blank with WST-1 reagent carried. After two hours of incubation at above culture conditions, the well plate was shaken for a minute, and light absorption was measured in the ELISA reader at a wavelength of 420 nm. The reference was determined at a wavelength of 600 nm. The use of cells was approved by the Ethics Committee of the Saarland Medical Association (E168/09).

## 3. Results

### 3.1. Fluorescence Microscopy

Overall, attachment areas of fibroblasts on machined titanium were significantly larger after 4 h, 12 h and 24 h of cultivation (Figure 1 and Figure 2). During the initial colonization (4 h), this was significant for both machined and microstructured (Figure 3 and Figure 4) surfaces. The cell count was higher in treated samples for both the machined and microstructured titanium during the entire period (Figure 5 and Figure 6).

### 3.2. Scanning Electron Microscopy

After 4 h of culture, the fibroblasts on the machined and microstructured titanium discs showed a dense and homogeneous colonization pattern. The cell bodies of the fibroblasts on the plasma-treated machined surfaces showed a more elongated configuration with a greater and stronger attachment to the surface and stronger intercellular contacts compared to the controls without plasma treatment (Figure 7). On the microstructured titanium, the cells displayed no clear differences in their phenotypes on plasma-treated and untreated titanium. However, the attachments of the fibroblasts to plasma-treated titanium appeared stronger (Figure 7 and Figure 8). After 12 h, the results were comparable to those observed after 4 h of culture, and after 24 h, the results were inconclusive because the fibroblasts detached from some titanium surfaces during SEM preparation.

### 3.3. Biological Activity

The analysis of the biological activity showed no negative effect of the plasma treatment on cell proliferation over the cultivation periods of 4 h to 24 h. However, no clear beneficial effect was observed by plasma treatment either. The results for alkaline phosphatase activity, cell proliferation assay and the measurement of nucleus size are presented in the Appendix Materials (Figure A1, Figure A2, Figure A3, Figure A4, Figure A5 and Figure A6 → Appendix A, Appendix B, Appendix C, Appendix D, Appendix E and Appendix F).

## 4. Discussion

To the best of the authors’ knowledge, this is the first study demonstrating the beneficial effects of the conditioning of microstructured or machined titanium with cold atmospheric plasma on the initial attachment of primary human gingival fibroblasts.

It has already been demonstrated in several studies that osteoblasts respond positively to the treatment of titanium surfaces with cold atmospheric plasma [14].

In this in vitro study, primary human gingival fibroblasts were investigated. They are the main cells of peri-implant mucosa surrounding the implant. Tight and inflammation-free enclosure of the implant is essential for long-term implant success [20].

It has been demonstrated that the treatment of machined and microstructured titanium surfaces with cold atmospheric plasma results in more homogeneous and more intense initial adhesion of gingival fibroblasts. This is discussed as an advantage for the re-osseointegration of titanium implants [21,22].

To analyze the initial colonization of plasma-treated surfaces, primary human gingival fibroblasts of the sixth passage were used. These cells allow a closer approach to the real situation; nevertheless, they are more difficult to culture than immortalized cells. The titanium test specimens used were made of medical-grade pure titanium (titanium grade 2) with different surface configurations. Both microstructured surfaces, which were sandblasted and etched, and machined titanium specimens were used to match the implant structure of common implants, which have a machined and a microstructured part [23]. The treatment was performed using a miniaturized plasma source, which has been used and characterized several times [24,25]. The experiments were performed under ambient conditions. Care was taken to select parameters guaranteeing biologically acceptable temperatures in the treatment area. Thus, the closest possible approximation to realistic conditions in practice was simulated. The cultivation period was limited to a maximum of 24 h, so that the prophylactic addition of antibiotics to the medium could be avoided. The analytical methods used were aimed at visualizing the attachment, morphology and biological activity of fibroblasts.

Both plasma-treated and non-treated control samples showed cell proliferation up to 24 h. The scanning electron microscopy analysis of the microstructured samples showed that the fibroblasts grew into the structure of the titanium sample surface. This confirms the results of older studies underlining the excellent biocompatibility of titanium [26,27].

Overall, the plasma-treated titanium surfaces showed benefits especially for initial colonization. After 4 h of cultivation, slight advantages were shown for all investigated parameters, such as cell attachment, cell number, metabolic activity and nucleus size. After a longer culture period, these advantages were not as distinct. The results of the analysis of cell proliferation (WST-1) showed no clear advantage for plasma treatment. For the intensity of initial attachment as the most important parameter, treatment with cold atmospheric plasma showed a clear improvement over the entire analysis period [28,29].

Adhesion is a multistep process [30]. Firstly, the titanium surface encounters an aqueous medium in which an electrochemical bilayer forms [31], possibly modified by plasma treatment. The surrounding medium also contains proteins such as fibronectin or growth factors helping the cell adhere to the surface. The proteins absorb at the surface [32]. After the first cells have adhered to the surface, an extracellular matrix (ECM) is established. The ECM is the basis for a firm adhesion of cells on a surface, and its structural basis is formed by integrins. For dental practice, the results of this study may be of importance. Treatment of titanium with cold atmospheric plasma improves the wettability and thus the distribution of applied biological substances and the cells on surfaces. This is in line with other findings [22]. The improvement in initial attachment demonstrated in this study provides the opportunity for use of cold atmospheric plasmas in implantology, both in primary implant placement and in the regeneration of lost attachment due to peri-implantitis [33,34]. Additionally, cold atmospheric plasmas can be used for the disintegration of biofilms on implant surfaces [35].

Further studies are necessary to assess the potential of cold atmospheric plasma in dentistry. However, there is no question that its application offers interesting possibilities, especially for the re-osseointegration of titanium implants.

## 5. Conclusions

The present in vitro study demonstrated the beneficial effect of plasma treatment on the initial attachment of primary human gingival fibroblasts to titanium. This was observed for both the machined and the microstructured titanium surfaces. The results support considerations for the clinical application of cold atmospheric plasma in the context of pre-implantation/peri-implantitis treatment.

## Figures and Tables

**Figure 1 biomedicines-11-01185-f001:**
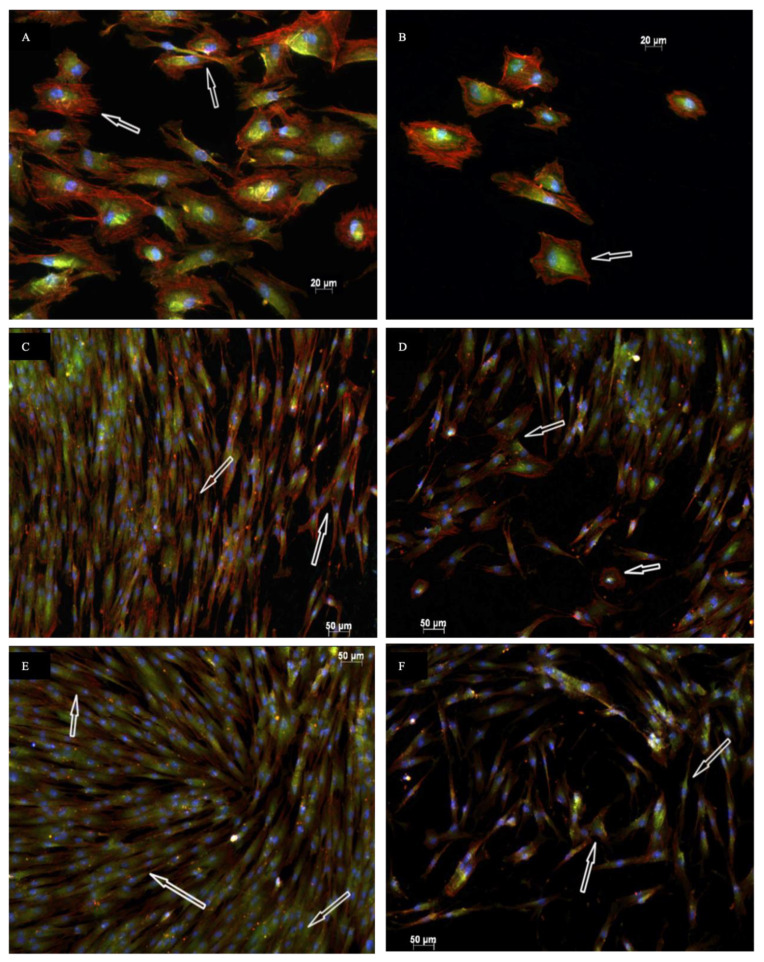
Representative fluorescence microscopic images of fibroblasts on machined titanium discs after 4, 12 and 24 h of cultivation (Vinculin, Phalloidin, DAPI stains). (**A**) Plasma-treated sample after 4 h of culture. Cells show planar attachment of the individual cells (arrow: green staining, vinculin). (**B**) Control sample, after 4 h of culture with rounded cell bodies (arrow) with small areas of adhesion (green). (**C**) Treated sample after 12 h of culture. Arrows: elongated cells with intensive attachment and cell interaction (green). (**D**) Control sample, after 12 h. Arrows: more round cells are present and less attachment (green). (**E**) Treated sample after 24 h. Arrow: intensive attachment of cells, elongated configuration. (**F**) Control sample after 24 h of culture. Arrow: less cells and lower focal attachment (green) of the individual fibroblasts in comparison to the treated samples (**E**).

**Figure 2 biomedicines-11-01185-f002:**
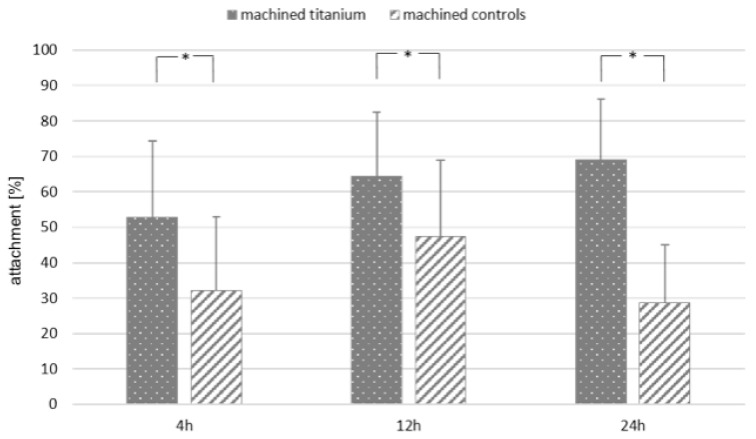
Comparison of relative proportion of focal adhesion (Vinculin) after 4, 12 and 24 h of fibroblast cultivation on machined titanium probes with controls. The bars correspond to the mean value, and the line on top corresponds to the ±standard deviation. Statistically significant differences (*p* < 0.05) are marked with an asterisk.

**Figure 3 biomedicines-11-01185-f003:**
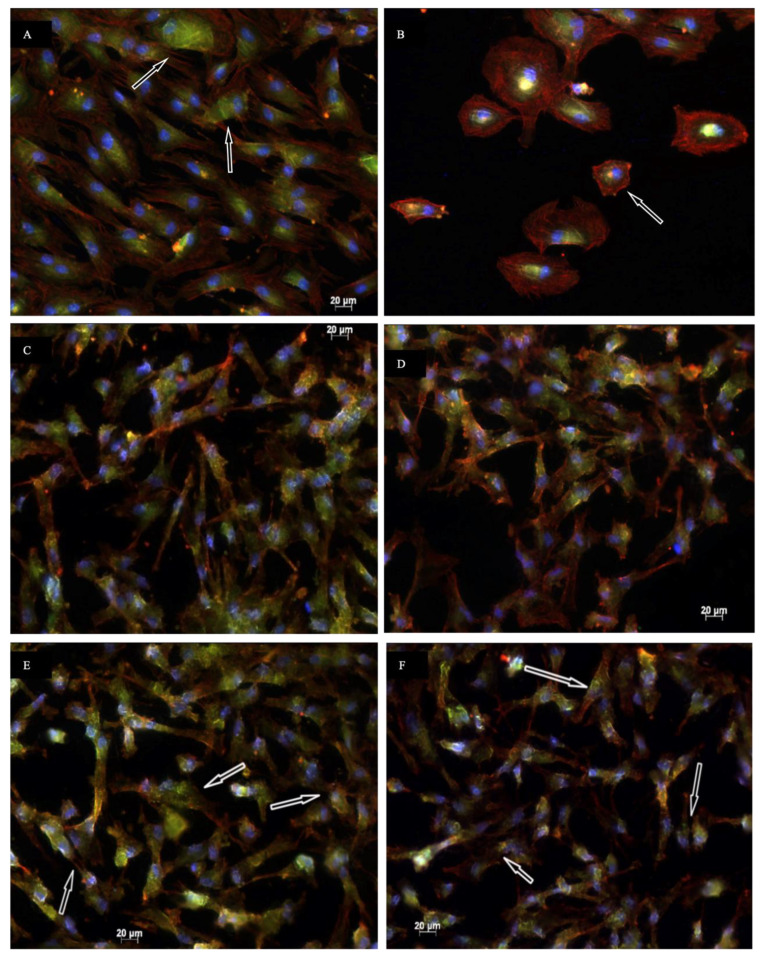
Representative fluorescence microscopic images of fibroblasts on microstructured titanium discs after 4, 12 and 24 h of cultivation (Vinculin, Phalloidin and DAPI stains). (**A**) Plasma-treated titanium after 4 h of culture. Cells show larger adhesion areas (green, arrows) in comparison to (**B**) control sample, gas stream without plasma ignition (red, arrow). (**C**) Treated and (**D**) untreated titanium discs after 12 h of culture. No differences are visible for the attachment of the fibroblasts. (**E**) Treated sample after 24 h of culture. Arrows: minimally more cells and stronger attachment of the individual cells and larger cell bodies in comparison to (**F**) untreated titanium discs.

**Figure 4 biomedicines-11-01185-f004:**
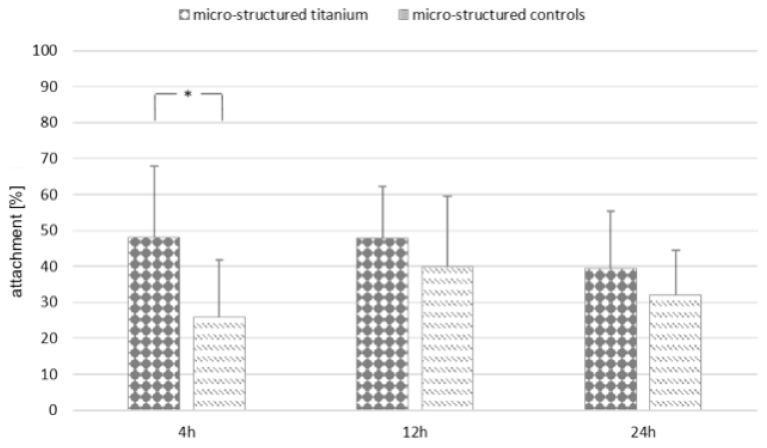
Comparison of relative proportion of focal adhesion (Vinculin) after 4, 12 and 24 h of fibroblast cultivation on microstructured titanium probes with controls. The bars correspond to the mean value, and the line on top corresponds to the ±standard deviation. Statistically significant differences (*p* < 0.05) are marked with an asterisk.

**Figure 5 biomedicines-11-01185-f005:**
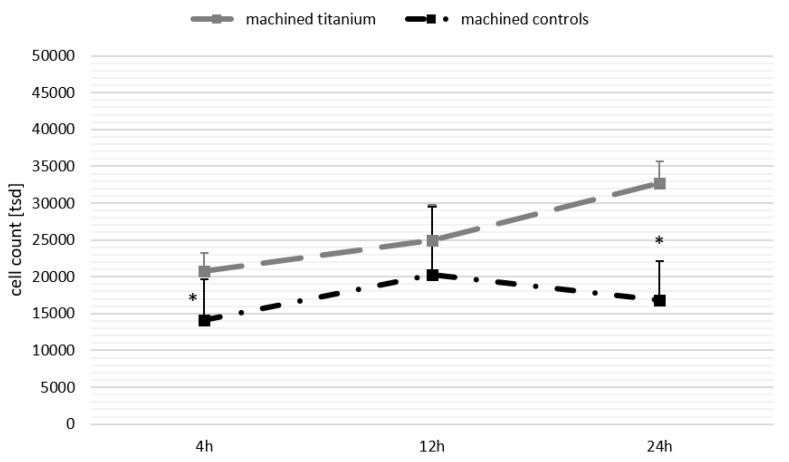
Comparison of cell count after 4, 12 and 24 h between machined titanium samples and controls. The squares correspond to the mean value, and the line on top corresponds to the ±standard deviation. Statistically significant differences (*p* < 0.05) are marked with an asterisk.

**Figure 6 biomedicines-11-01185-f006:**
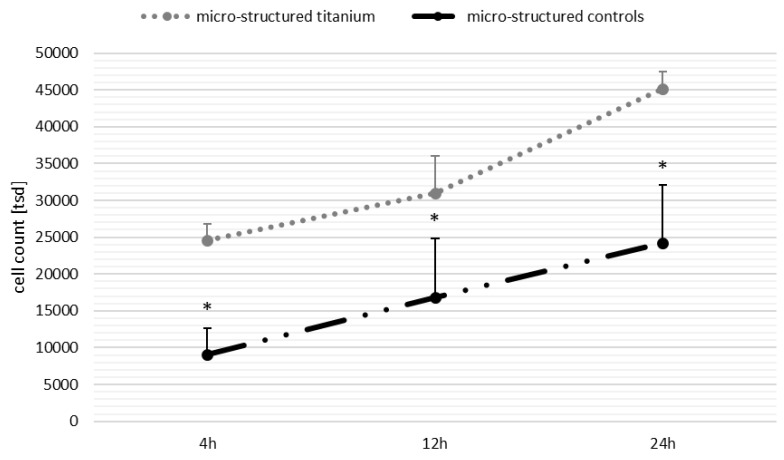
Comparison of cell count after 4, 12 and 24 h between microstructured titanium samples and controls. The circles correspond to the mean value, and the line on top corresponds to the ±standard deviation. Statistically significant differences (*p* < 0.05) are marked with an asterisk.

**Figure 7 biomedicines-11-01185-f007:**
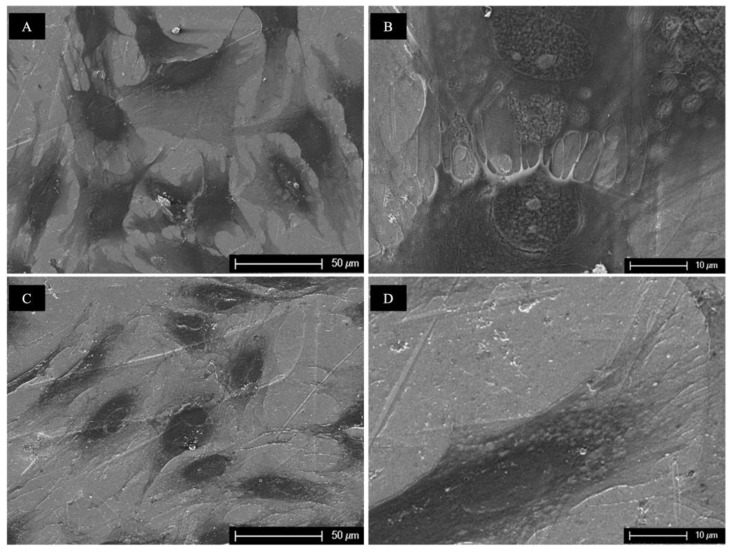
Representative scanning electron microscopy images of fibroblasts on machined titanium discs after 4 h of cultivation. (**A**,**B**) Plasma-treated samples. Fibroblasts are larger and show more intensive attachment and intercellular contacts in comparison to (**C**,**D**) controls without plasma treatment.

**Figure 8 biomedicines-11-01185-f008:**
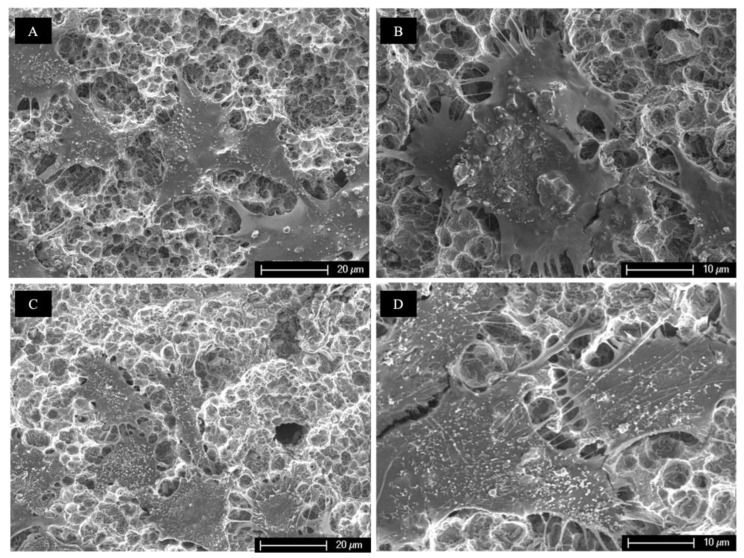
Representative scanning electron microscopy images of fibroblasts on microstructured titanium discs after 4 h of cultivation. (**A**,**B**) Plasma-treated samples. (**C**,**D**) Controls without plasma treatment. Fibroblasts show intensive intercellular contacts and attachment to the titanium, suggesting favorable conditions for cell–surface interaction.

## Data Availability

The data supporting the findings of this study are available from the corresponding author upon reasonable request.

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
