# Peer review of "Effects of Cold Atmospheric Plasma Pre-Treatment of Titanium on the Biological Activity of Primary Human Gingival Fibroblasts"

_biomedicines, 2023, doi:10.3390/biomedicines11041185_

Round 1

Reviewer 1 Report

Dear Authors,

The manuscript biomedicines-2284656, entitled 'Effects of cold atmospheric plasma pre-treatment of titanium on the biological activity of primary human gingival fibroblasts' presents results on the effects of cold plasma treatment on the titanium surfaces, with interest on the biological effects upon fibroblasts.

The overall manuscript is well written and structured, with good examples and well discussed results. The figures ar of good quality and well-presented and discussed.

I would, however, include in the material section also some info, schematics, references on the plasma source used in these experiments. It is a microware plasma source, but it is related to the micro plasma source from a COST program? What about the plasma characteristics and diagnosis? references to that? maybe a current-voltage, Lissajous, or even some OES data (figure) would be of immense help for the reader. The temperature of the plasma, the RONS and spreading of the plasma plume onto the titanium surface also are to be of great help, so it should be documented in the manuscript.

ACCEPT after considering the above suggestions.

Author Response

Please see the rebuttal letter.

Reviewer 2 Report

The adhesion and biological activity of primary human gingival fibroblasts on titanium that has undergone cold atmospheric plasma (CAP) treatment have been studied by the authors. Although this effort is unique, the results are not satisfactory. It is therefore encouraged to thoroughly read the comments and suggestions before submitting the updated manuscript.

A. The results of Figure 4 are in contradiction to Figure 3, which depicts the green (vinculin) and cell number growing with time. Can the authors clarify this or explain the conflicting study results?

B. The y-axis in Figures 2 and 4 has a spelling error; it can potentially be changed to Cell attachment (%).

C. Comparisons of the cell populations in Figures 1E and 3E show that machine titanium has more cells than microstructure, although being smaller in size. Figures 5 and 6 though presented different findings.

D. The data given under SEM and the cell morphology are unsatisfactory. For the cell morphology, CLSM images would be ideal.

E. Lines 118–119, the authors stated, "These images were used to analyze the number of cells, the attachment behavior, and the average nucleus size." 1. Could you please rephrase the statement? 2. Using ImageJ or another piece of software, explicitly explain how the calculation was done to get at the cells' quantitative behavior.

F: Is it colonisation or colonization? Please just use one word throughout the entire manuscript.

H. Please cite lines 265-266, 269–274 in the literature to support these findings.

H. Lines 29-30, and 243-249, references needed

I. Figures 9 - 14 in the supporting information need to be numbered appropriately, and why the cell proliferation is higher in the control group and declines over time in the samples?

J. Could you please begin the conclusion with some better wording, repeating this for begining of the two consecutive sentences is not ideal.

K. There have been some work on Cold atmospheric plasma treatment (CAP) of titanium implants over the past years, please cite some recent references form 2021-2023.

L. Please remove the template text on lines 155-157.

Author Response

Please see the rebuttal letter.

Reviewer 3 Report

Dear authors!

The manuscript „Effects of a cold atmospheric plasma pre-treatment of titanium on the biological activity of primary human gingival fibroblasts” monitors the effects of an atmospheric plasma treatment with a gas mixture of the noble gas helium and oxygen. However, the work focuses on the enhanced focal adhesion of human gingival fibroblasts to plasma-treated titanium surfaces in comparison to untreated references.

A few general remarks:

What is exactly the difference between the machined and microstructured titanium? Is it possible to show the reader a picture? Do you have micrograph or a sketch of the structures? Alternatively, is it just a random structure giving you a certain roughness (here: 2µm, Lit: Lampin: Correlation between substratum roughness and wettability, cell adhesion, and cell migration), micrograph? Several authors see a difference of micro- and macrostructured surfaces (e.g., Elter et al.: The influence of topographic microstructures on the initial adhesion of L929 fibroblasts studied by single-cell force spectroscopy). What are the dimensions of your structure? Clearly spoken, the dimensions of the structure are the basis to distinct the effects of the structure on the cells or, in part, the proteins building up the ECM. Did the plasma treatment possibly change the roughness of your titanium? Scan [14] more carefully.

Please describe titanium, at least its surface, a bit more in detail. Titanium possesses a passivation on its surface, which protects it from corrosion. Either the passivation is an auto passivation or it is processed (e.g., chromate conversion coating). Does it alter through a plasma treatment with an oxidizing media? The mixture, especially when oxygen is used, is described for its etching abilities (Cvelbar et al.: Selective oxygen plasma etching of coatings). Is a possible chemical change a drawback or a benefit for the cell adhesion? Does the surface charge change, i.e. do you introduce new charges through an oxygen-plasma treatment. Did you measure? (Changes of the contact angle. If not, please discuss. You will find a lot of literature (e.g., Shin: Treatment of Metal Surface by Atmospheric Microwave Plasma Jet). In general, what does the literature say?

Adhesion is a complex process, which can be described in terms of a succession (Gongadze et al: Adhesion of osteoblasts to a nanorough titanium implant surface). A brief overview: First step, your titanium surfaces encounters an aqueous medium, where an electrochemical double layer builds up (see Israelachvili, Iglic), which can be altered by a plasma treatment, too. The surrounding media also possesses proteins like fibronectin or growth factors that helps the cell to adhere at the surface. The proteins absorb to the surface (JVC Neto et al.: Protein absorption on titanium surfaces treated with a high-power laser: A systematic review). Subsequently, after the first cells adhered to the surface, an extracellular matrix is established. The ECM is the basis for a tight connection of cells to a surface and the structural motives of it can be sensed by integriens. You know! Did you check the literature of various authors that investigate cell adhesion with optical tweezers (Schwingel) or single cell force spectroscopy (SCFS, Taubenberger, Elter, DJ Müller)? How did you monitor the strength of adhesion? As an argument for a high coverage of the surface, through the expression of several proteins, etc.? Please explain in detail. Against that background, please describe the function of vinculin and its relationship to focal adhesion in some detail. Underpin it with literature.

Line 47: Please check the literature; check my chapter about adhesion above. In general, please describe known facts of your experimental design (physical behavior of the surface, biological behavior of cells in contact with that surface,…) in the Introduction (when you just giving facts) or in the discussion (if they are needed to classify your results).

Line 87: I don´t understand the phrasing 10 4?

Figure 2: Think about your statistics. You made three t-tests. Why do we see a p-value smaller than 0.05 for all time points. Is it the alpha (a = 0.05), which have to undercut to reject your null hypothesis or do you see a greater significance for the values after a 24h growth of cells on the treated surfaces?  

Please write a more conscientious discussion. What do other authors say (SCFS, optical tweezers). What are the effects of various working gasses of the plasma source?

Author Response

Please see the rebuttal letter.

Round 2

Reviewer 2 Report

Authors have efficiently responded to all the comments. 

Author Response

Please see the rebuttal letter.

Reviewer 3 Report

In line 80 and 91 are some digits, which should be subscripted.
